# Tuning the Bioactive Properties of *Dunaliella salina* Water Extracts by Ultrasound-Assisted Extraction

**DOI:** 10.3390/md21090472

**Published:** 2023-08-27

**Authors:** Joana P. A. Ferreira, Madalena Grácio, Isabel Sousa, António Pagarete, M. Cristiana Nunes, Anabela Raymundo

**Affiliations:** 1LEAF—Linking Landscape, Environment, Agriculture and Food—Research Center, Associate Laboratory TERRA, Instituto Superior de Agronomia, Universidade de Lisboa, Tapada da Ajuda, 1349-017 Lisboa, Portugal; madalenagracio@outlook.com (M.G.); isabelsousa@isa.ulisboa.pt (I.S.); crnunes@gmail.com (M.C.N.); anabraymundo@isa.ulisboa.pt (A.R.); 2Pagarete Microalgae Solutions Soc. Unipessoal Lda., Rua João Chagas, 4, 7Esq, 1495-069 Algés, Portugal; pagarete@gmail.com

**Keywords:** *Dunaliella salina*, ultrasound, extraction, bioactivities, sensory analysis

## Abstract

(1) Background: Microalgae are promising feedstock for obtaining valuable bioactive compounds. To facilitate the release of these important biomolecules from microalgae, effective cell disruption is usually necessary, where the use of ultrasound has achieved considerable popularity as an alternative to conventional methods. (2) Methods: This paper aims to evaluate the use of ultrasound technology in water medium as a green technology to recover high added-value compounds from *Dunaliella salina* and improve its sensory profile towards a high level of incorporation into novel food products. (3) Results: Among the variables, the solid concentration and extraction time have the most significant impact on the process. For the extraction of protein, or fat, the most influential factor is the extraction time. Total polyphenols are only significantly affected by the extraction time. The antioxidant capacity is strongly affected by the solid to liquid ratio and, to a small extent, by the extraction time. Ultrasound-assisted extraction improves the overall odor/aroma of *D. salina* with good acceptability by the panelists. (4) Conclusions: The application of ultrasonic-assisted extraction demonstrates a positive overall effect on enhancing the sensory profile, particularly the odor of microalgal biomass, while the bioactive properties are preserved. Notably, the intense sea/fish odors are reduced, while earthy and citrus notes become more prominent, resulting in an improved overall sensory profile score. This is the first time, to our knowledge, that this innovative, green, and efficient technology has been used to upgrade the aroma profile of microalgae.

## 1. Introduction

In the ever-evolving world of consumer preferences, a notable shift is taking place towards products that embody natural goodness, promote well-being, and boast transparent labels—a phenomenon often referred to as the “Food Trends of 2023” (https://www.innovamarketinsights.com, accessed on 25 August 2023). This ongoing transition is not only understandable but also inspiring, as individuals increasingly seek out items that align with their health-conscious and eco-friendly lifestyles. Among the myriad options that cater to these evolving sensibilities, microalgae have emerged as a frontrunner, carving out an essential niche in the realm of food products [1].

What distinguishes microalgae from the crowd is their multifaceted role in direct food consumption, which stems from their innate health benefits, natural pigments, and their capacity to serve as an exceptional vegan source of protein. This trifecta of attributes has propelled microalgae into the spotlight, capturing the attention of health-savvy individuals and eco-conscious consumers alike [2].

Delving into the essence of microalgae, it becomes apparent that they are more than just microscopic organisms—they are nutrient-packed powerhouses. They house a treasure trove of health-promoting compounds, such as proteins, polysaccharides, and an array of bioactives, including vitamins, minerals, antioxidants, and anti-inflammatory agents. What is even more remarkable is that these nutritional gems are naturally ensconced within the confines of the microalgae cells. This natural encapsulation mechanism is a key factor in considering microalgae as clean label ingredients, which aligns harmoniously with the contemporary consumer ethos. Microalgae seamlessly assume this role, providing a clean label option brimming with bioavailable nutrients [3].

Consumers are adopting healthier lifestyles and have been increasingly interested in the intake of products with health-promoting properties, such as microalgae [2], and their beneficial properties are attributed to the presence of biologically active molecules. Many of these beneficial compounds are stored within microalgae cells, requiring the rupture of the cell wall for their extraction and recovery. The controlled disruption of the cell wall plays a crucial role in maximizing the bioavailability of these bioactive compounds. The first step in the recovery of bioactive compounds from microalgae involves an effective cell disruption process, by means of conventional methodologies, such as maceration and Soxhlet extraction, or of innovative technologies, such as ultrasound-assisted extraction (UAE). Because bioactive compounds are sensitive to extraction techniques based on heat or solvent use, ultrasound has gained tremendous interest in the research. It is a green extraction technique, safe, sustainable, non-thermal, with inexpensive procedures, reducing the requirements in terms of solvents, energy, and time and the production of hazardous substances [4]. Different authors have already reported the extraction of added-value compounds from microalgae species (e.g., *Nannochloropsis* spp., *Spirulina* spp., and *Chlorella* spp.). Phenolic compounds, chlorophylls, fatty acids, and proteins were efficiently recovered from *Nannochloropsis* spp., *Arthrospira platensis*, and *Chlorella vulgaris* microalgae, respectively [5,6]. If, on the one hand, microalgae are becoming more relevant in the food market because consumers have greater awareness of nutrition [5], on the another hand, their incorporation with foods has some major drawbacks in terms of the sensory properties and consumer acceptance, mainly due to the aroma, taste, and coloration [6]. Several technological developments have been attempted to improve the sensory properties of microalgae biomass, involving random DNA alteration/controlled DNA manipulation, the adjustment of growth conditions, downstream processing, micro-encapsulation, or bioprocessing [6]. Extraction methodologies are being used to improve the sensory attributes [6]; however, it is essential to focus on clean technologies, and again, UAE can play an important role here. 

Over the last two decades, several microalgae species have been investigated and exploited as an emerging source of bioactive compounds, widely applied in the fields of nutraceuticals, pharmaceuticals, and cosmetics, and have attracted considerable attention in recent years [1]. *Dunaliella* sp. stands out as a remarkable microalga with the capacity to accumulate significant amounts of β-carotene, ranging from 8% to 14% of its total dry weight under suitable conditions. The array of compounds produced by *Dunaliella* sp., including carotenoids (especially β-carotene), glycerol, and cosmetic compounds, as well as its biomass, holds substantial industrial significance as a valuable protein source [7]. The lyophilized or dried form of *Dunaliella* biomass, along with its carotenoids, finds practical application in food and feed industries as additives for coloring or as supplements for both human and animal consumption [7].

This work is integrated in a European project funded by the EEAGrants, de DoMar—Development of Microalgae Advanced Resources Project. The project main goal is to implement and develop in Portugal innovative and sustainable technologies for the production of high-quality low-cost microalgal biomass. Pagarete Microalgae solutions, the project promotor, uses a highly advanced technology developed in partnership with the German company Omega Green for the cultivation of microalgae with more immediate interest in the aquaculture sector: *Nannochloropsis, Tetraselmis*, and *Dunaliella*. Since *Dunaliella* is already used in feed products and is under evaluation by the FDA for food consumption, it makes a lot of sense to perform thorough studies to understand the nutritional, chemical, and biochemical compositions and its sensory profile, and how we can improve its quality for both feed and future food applications. This work focuses on the use of UAE to improve the sensory qualities of *Dunaliella salina* biomass using water as a medium, and obtain aqueous extracts nutritionally relevant and enriched with bioactive compounds for the further development of novel and healthy food products.

## 2. Results and Discussion

### 2.1. Impact of Ultrasound-Assisted Extraction Parameters

#### 2.1.1. On the Yield

To understand the advantages of using UAE for the recovery of added-value extracts, a first batch of *D. salina* powder was extracted firstly using a Soxhlet extractor for 18 h and a second batch was also extracted by maceration over 24 h, also in water. The correspondent extracts were obtained with 10.4% and 3.1%, respectively. In this study, 1:10 and 1:5 of solid to solvent ratios (or 10% and 20% of solids) were used, maintaining the solvent (water) volume at 500 mL. The minimum and maximum yields obtained by UAE were 14.7% (pulse supply mode, 30 min, r.t., 1:5 sample to solvent ratio) and 41.4% (continuous, 30 min, r.t., 1:10 sample to solvent ratio), respectively (Table 1). Compared to the conventional methods, UAE allowed obtaining extracts in higher yields (10% with Soxhlet and 3% with maceration, vs. 15–41% with UAE), supporting the idea that UAE is an efficient and sustainable technique to recover added-value compounds in a shorter time (10–30 min using UAE and 18–24 h using the conventional methods) and with a reduced use of the solvent.

The most significant factors affecting the yield in UAE were the solid to solvent ratio and the duration of the extraction, as can be seen in Table 1. The higher yields of extraction were obtained when using a 1:10 solid to solvent ratio (almost 2-fold), either in continuous or pulse modes. To analyze the relationship between the yield of extraction and the solid to solvent ratio and extraction time, we can compare the values within each column for continuous and pulse modes separately. For the continuous mode, at a solid to solvent ratio of 1:10, the yield of extraction increased with the time (from 35.8% to 41.4%). At a solid to solvent ratio of 1:5, the yield of extraction decreased to some extent with the time, from 10 to 20 min (from 23.5% to 15.8%). At 30 min, the yield somewhat increased (17.5%). In the pulse mode, and at a solid to solvent ratio of 1:10, the yield of extraction generally increased with time. At 10 min, the yield of extraction was 22.8%, which increased to 33.4% at 20 min and slightly further to 35.3% at 30 min. However, at a solid to solvent ratio of 1:5, the yield of extraction remained relatively constant with the time (around 14.7%). In summary, for the continuous mode, the yield tended to increase with time at a solid to solvent ratio of 1:10, while it decreased with time at a solid to solvent ratio of 1:5. In the pulse mode, the yield also increased with time at a solid to solvent ratio of 1:10, while it remained relatively constant at a solid to solvent ratio of 1:5. In general, microalgae and plant cells are disrupted more by longer extraction times and, consequently, the release and diffusion of the bioactive compounds are enhanced [8,9]. The solid to solvent ratio is also a very important parameter for the extraction process. According to Purohit and Gogate [10], the use of a lower solid to solvent ratio than the optimum value leads to an increase in the solvent consumption and higher solid to solvent ratios than the optimum value will result in an incomplete extraction. Using a solid to solvent ratio of 1:10, the yield of the extraction was higher than when using a 1:5 solid to solvent ratio, and this could be explained by the excessive amount of microalgal material (1:5 ratio), which caused the increase in viscosity, and thus inhibited the diffusion of compounds through the extraction medium [11]. In addition, it contributed to ultrasound wave attenuation, leaving the restricted zone located near the ultrasound probe as the active part [12].

The use of pulsed UAE also negatively influenced the extraction yield, either using 1:10 or 1:5 solid to solvent ratios. In fact, in some cases, it was possible to observe a lower extraction yield when using a pulsed ultrasound compared to the continuous mode [13,14]. Several factors can contribute to this outcome, such as the energy distribution that may result in an uneven distribution and dissipation of energy within the extraction medium, leading to a lower overall yield [15]. In addition, continuous cavitation effects, such as the formation of small bubbles or voids within the liquid medium, enhance the extraction process [16]. Mass transfer limitations can also explain why the extraction yields were higher when using the continuous sonication mode, since it provided the continuous agitation and disruption of the extraction medium, facilitating mass transfer and improving the extraction efficiency [14,15].

#### 2.1.2. Protein Content

Figure 1 shows the protein content (%) in biomass and extract fractions obtained through continuous and pulsed ultrasound-assisted extractions at different time intervals and solid to solvent ratios. When comparing the different UAE conditions, the protein content was almost 3-fold higher in biomass fractions than in the extracts (for example, 28.3% and 8.7%, respectively, when using a 1:10 solvent ratio after 30 min in the pulsed sonication mode). Additionally, when using a solid to solvent ratio of 1:10, the protein content in the biomass, as well as in the extract fractions, was slightly higher than with the 1:5 ration, which was the opposite of what was expected. The extraction process usually becomes more energy efficient at a higher algal biomass concentration because it is more effective in its wave contact with solid matter, such as microalgae cells, as for all released components. However, in some cases, when an excessive amount of algal biomass is present, the diffusion of compounds towards the extraction medium becomes difficult due to an increase in viscosity, as previously mentioned [11].

Based on these data, it is difficult to draw definitive conclusions about the extraction efficiency or the impact of continuous versus pulsed ultrasound-assisted extraction and time of extraction on the protein yield. It appears that the protein content varies slightly between the different extraction conditions; however, the differences are not substantial. When compared to the extractions with conventional techniques, the extracts obtained by UAE showed similar protein contents than the ones obtained from the Soxhlet extraction (around 8.5% dw).

#### 2.1.3. Fat Content

As for the other intracellular constituents, the different ultrasound extraction parameters influenced, to different extends, lipid extraction yields. When analyzing Figure 2, is evident that the biomass fractions have a higher fat content (varying between 11.2% and 17.5%) than the correspondent extract fractions (0.5% to 6.8%), which means that the selected conditions are not efficient to remove lipids from within the microalgae cell. In fact, the use of polar solvents as water is not effective for the extraction of lipids. Ranjan and co-workers [16] claimed that solvent selectivity is usually the most effective parameter concerning the degree of lipid extraction. In a study conducted by Mecozzi and co-workers [17], it was confirmed that sonication with diethyl ether resulted in higher lipid extraction yields from marine mucilage compared to sonication with methanol. Another study by Wiyarno and co-workers [18] focused on the UAE of algal lipids from *Nannochloropsis* sp., highlighting the influence of different solvents on the efficiency of ultrasonic extraction. It was observed that when ethanol was used, higher extraction temperatures and longer extraction times were required compared to the use of n-hexane, suggesting that the selection of the solvent is an important factor for optimizing the UAE of algal lipids.

The extraction time is also an important factor for lipid extraction [19]. Generally, by increasing the sonication treatment time, cell disruption occurs, as well as an increase in the amount of released intracellular constituents [20]. However, after an optimal sonication time, usually no significant differences are noticed, suggesting that a short sonication time is enough to obtain a suitable yield [21]. When analyzing Figure 2, it is quite evident that the lipid extraction yields increase with the increasing sonication time (almost 4- and 6-fold when using solid to solvent ratios of 1:10 and 1:5, respectively). It is also worth noticing that longer extraction times lead to increases in the temperature and vapor pressure, promoting the formation of many cavitation bubbles. They collapse with less intensity due to the reduced pressure difference between the inside and outside of the bubbles [22], thus reducing the intensity of the mass transfer enhancement. Considering the solid to solvent ratio, it is possible to see that another important factor for the lipid extraction process is that different solid to solvent ratios, i.e., 1:10 and 1:5, result in the different recovery rates of the lipids: the highest lipid recovery rate was observed when the solid to solvent ratio was 1:5 *w*/*v*. It is expected to see a more efficient extraction process at a higher algal biomass concentration because it is more effective in the wave contact with solid matter, such as microalgae cells, as for all released components [16]. When compared to the results of the extracts obtained from the Soxhlet extraction, it is worth noticing that a lower fat content is obtained.

#### 2.1.4. Ash

The extraction of minerals from microalgae using ultrasound has gained attention in recent years. Microalgae are extremely rich sources of minerals, namely, calcium, magnesium, iron, and trace elements, such as selenium and zinc. The cavitation and mechanical forces generated by ultrasound promote the rupture of cell membranes, aiding the liberation of minerals from the microalgal biomass [12].

When analyzing Figure 3, is evident that the biomass fractions have a higher mineral content (varying between 33.2% and 57.5%) than the correspondent biomass fractions (7.3% to 9.1%), which means that, in the selected conditions, the minerals are efficiently entirely removed from the cell towards the liquid phase. The extraction time seems to be an important factor for mineral recovery [19]. Usually, by increasing the sonication treatment time, we increase the amount of released intracellular constituents [20]. In fact, there is an increase in minerals that are released for the extracts over time (with the exception of the following extraction conditions: a pulse sonication mode of 20 min and a solid to solvent ratio of 1:10). Overall, the solid to solvent ratio and sonication mode had a significant effect on the mineral recovery rate, especially in the pulse sonication mode when the solid to liquid ratio increased (around 1.5-fold).

Additionally, the extracts obtained by different UAE conditions showed a higher ash content when compared with the ones obtained using conventional techniques.

#### 2.1.5. Carbohydrates

The effect of the sonication time, solid to solvent ratio, and sonication mode were analyzed. Figure 4 shows that the total carbohydrate content increases with the extension of the extraction time, except after 20 min in the pulsed sonication mode and when the solid to liquid ratio is 1:10. This indicates that the extraction time can improve the efficiency of carbohydrate extraction, which is probably due to the fact that algal cells can break more effectively in these conditions [21,22,23]. The solid to solvent ratio also affected the extraction efficiency: the carbohydrate content was higher in the extracts than in the biomass, which was more evident when the solid to solvent ratio was 1:5, either in continuous or pulse sonication modes. The same trend was reported by Zhao and co-workers [23]; however, it was the opposite of what we expected because the excessive amount of microalgal material (1:5 ratio) usually inhibited the diffusion of compounds through a more viscous extraction medium [11]. The sonication mode alone had no influence on the carbohydrate’s extraction yield. In order to fully extract the carbohydrates, a combination of different UAEs could be tested as a low power input combined with longer extraction times [23] or combining UAE with other disruption methods, such as ozonation, microwave, homogenization, or enzymatic lysis to facilitate the release of the target compounds [15].

UAE was efficient in the recovery of carbohydrates when compared with maceration; however, it was quite similar to Soxhlet extraction.

#### 2.1.6. Antioxidant Potential

*D. salina* can produce several compounds, including pigments such as α-carotene, lutein, and zeaxanthin [24]; polyphenols, such as phenolic acids, flavonoids, isoflavonoids, stilbenes, lignans, and phenolic polymers [25]; or phytosterols [26] with remarkable antioxidant properties. Due to their production of valuable bioactive ingredients, microalgae, such as *D. salina*, also represent promising opportunities in the field of functional foods and as food additives, since it is listed as having no known toxins, and GRAS and EFSA concluded that mixed β-carotenes obtained from algae as a food color is not of concern in relation to safety [27].

The antioxidant potential measured by DPPH (Figure 5a) ranged from 137.1–223.3 µmol Trolox/100 g dw for the biomass fractions, and between 385.0–414.7 µmol Trolox/100 g dw) for the extracts, which was almost 2-fold. The highest value observed for the extracts (414.7 µmol Trolox/100 g dw) was obtained under the following extraction conditions: 180 W, 100% amplitude, r.t. (24 °C), and a solid to solvent ratio of 1:10, in continuous sonication mode for 10 min (C 1:10 10’). The treated biomass with greater antioxidant potential measured by the DPPH assay (223.3 µmol Trolox/100 g dw) was obtained in the same conditions. Longer extraction times appeared to have a negative effect on the antioxidant potential of extracts, which could be related to an increase in the temperature on the reaction medium, which could lead to the degradation of thermostable compounds, as antioxidants.

However, we may conclude that, in the tested conditions, the extraction does not occur to its full extent, and it does not cause a complete cell wall disruption process, since there is a considerable amount of antioxidants in the biomass that are not released into the extracts. A possible solution might be enhancing the extraction time, maintaining a low temperature to avoid the degradation of thermostable compounds [15].

The FRAP values (Figure 5b) ranged from 62.6–87.9 mmol Trolox/100 g dw for the extract fractions, and were again almost 8-fold lower for the biomass (9.5–12.2 mmol Trolox/100 g dw). The highest value observed for the extracts (87.9 mmol Trolox/100 g dw) was obtained under the following extraction conditions: 180 W, 100% amplitude, r.t. (24 °C), and a solid to solvent ratio of 1:5, in the pulse sonication mode for 20 min (D 1:5 20’). The treated biomasses with greater antioxidant potential measured by the FRAP assay (11.9–12.2 mmol Trolox/100 g dw) were obtained using 180 W, 100% amplitude, r.t. (24 °C), and a solid to solvent ratio of 1:10, in the pulse sonication mode for 20–30 min. The observed results allow us to conclude that the UAE conditions allow the efficient extraction of compounds with antioxidant potential, which are determined by the FRAP assay towards the liquid phase, resulting in extracts with high antioxidant potential values, when compared to the biomass.

Except for the UAE, where a solid to solvent ratio of 1:5 in the pulse sonication mode was used, the antioxidant potential of the extracts decreased with the extraction time. Generally, lower extraction yields are obtained by prolonging the extraction time, because microalgae cells are disrupted to a greater extent with longer extraction times and, consequently, the release and diffusion of the bioactives are enhanced. However, when the extraction time is longer than the optimum time, the antioxidants might be degraded due to heat generation, resulting in the chemical breakdown of bioactive compounds and thereby decreases the extraction efficiency [15].

The antioxidant potential of extracts obtained from UAE was considerably higher when compared to the extracts obtained either by maceration (around 10 mg of Trolox/100 g dw) or Soxhlet (14 mg Trolox/100 g dw).

#### 2.1.7. Total Phenolic Content

A higher solid–solvent ratio usually facilitates improved solvent penetration into the microalgae cells, leading to the enhanced mass transfer of polyphenols and, consequently, an increased extraction yield. In fact, when the solid to solvent ratio was higher (1:5), the TPC of the extracts was almost 2.5-fold higher than when the solid to solvent ratio was 1:10 (ranging from 2.6–6.8 mg GA/100 g dw) (Figure 6). However, in some cases, excessively high amounts of plant material at smaller ratios can elevate solvent viscosity, hindering the diffusion of polyphenols through the extraction medium. This may explain why, when using the continuous sonication mode with a solid to solvent ratio of 1:10 after 30 min, a decrease in the TPC was observed (almost 2.6-fold). Or, when using the pulse sonication mode with a solid to solvent ratio of 1:10 and when using the continuous sonication mode with a solid to solvent ratio of 1:5, there was almost no variability in the TPC after 20 min. Moreover, it is worth noting that extending the extraction time may result in the oxidation of bioactive substances, potentially reducing the overall yield of phenolic compounds, which may justify what occurred after 30 min of extraction with a solid to solvent ratio of 1:10 in the continuous sonication mode (TPC decreased from 6.8 mg GA/100 g dw in C 1:10 20’ to 2.6 mg GA/100 g dw in C 1:10 30’). When comparing the effect of continuous or pulse sonication modes, an increase in TPC was observed, especially to a greater extent when using a solid to solvent ratio of 1:5. Christou et al. (2021) also reported an increase in the recovery of polyphenols by employing pulsed UAE [28].

In addition, it is important to notice that the TPC is usually closely related to the antioxidant potential, as reported by Ghafoor and co-workers [29]. However, greater differences in the TPC values between the different UAE conditions were observed when compared with the antioxidant potential. Usually, since both AAT and TPC are related to antioxidant activity, a substance with a higher total phenolic content is likely to exhibit stronger antioxidant activity. In many cases, a higher TPC value is indicative of a higher antioxidant capacity. However, it is essential to remember that while there is often a positive correlation between AAT and TPC, the relationship may not always be perfect. Some factors, such as the presence of other bioactive compounds or the specific chemical structure of the phenolic compounds, can influence the overall antioxidant activity of a substance, even if its TPC is high [30,31].

As for the antioxidant potential, the TPC of the extracts obtained from UAE was considerably higher when compared to the extracts obtained either from maceration (around 5 mg GAE/100 g dw) or Soxhlet (around 13 mg GAE/100 g dw).

### 2.2. Sensory Analysis

The sensory analysis assays were conducted with raw *D. salina* and biomass and extract fractions obtained after UAE. The panelist identified specific odors in the samples, ranging from floral, citrus, sea/fish, earthy, or none of the above. It was observed that *D. salina* raw microalgae had a very intense sea/fishy odor (identified by 45% of panelists) and earthy notes (identified by 30%), being less appreciated (Figure 7). After UAE, the biomass revealed a decrease in the intensity of the sea/fishy odor, being detected only by 10% to 25% of the panelists. Floral notes became more detectable (around 10% in raw *D. salina* and between 12% and 25% in the extracts). Citrus notes became detectable in the biomass (between 8% and 30% of the panelists) and were not detected either in the raw microalgae or in the extracts. In the extracts, the sea/fish odor was much more intense than in the raw microalgae (30–65% of panelist identified this specific and intense odor). Citrus notes were only detected in some extracts by a small number of panelists (10–20%). Earthy notes were detected in both the biomass and extract fractions, being much more intense than in the raw microalgae.

Figure 8 presents the average answers provided by the panel for the principal odors (citrus, sea/fish, and earthy) detected in the raw *D. salina* and the correspondent biomass and extract fractions obtained after UAE treatment. It is evident that the biomass is abundant in earthy notes, and a particular biomass fraction with citrus notes (obtained under the continuous sonication mode for 20 min and when the solid to solvent ratio was 1:5, C 1:5 20’). Extract fractions were enriched with sea/fish notes and also some earthy ones to a lesser extent.

Microalgae have some major drawbacks when we think of their sensory profile (color, odor, and aroma), especially if we want to incorporate them in feed and food products. There are several methods that can be used to improve these characteristics; however, they involve the use of solvents (chemical processes that involve the use of high amounts of solvents with a loss of bioactive properties) or enzymatic processes (expensive and not sustainable at the industrial scale). The use of a simple, practical, inexpensive, and green technology is efficient as the US represents a considerable contributor to this field of research. To our knowledge, no previous studies using this technology for the improvement of the sensory profile of microalgae have been found.

## 3. Methods and Materials

### 3.1. Samples and Chemicals

*D. salina* was produced by an autotrophic process by Pagarete Microalgae Solutions Soc. Unipessoal (Póvoa de Santa Iria, Portugal). The microalgal biomass was spray-dried by the same company and kept under −8 °C until further analysis.

Ultrapure water was obtained from the Synergy^®^ Water Purification System (Merck Millipore, Burlington, MA, USA). Methanol (suitable for HPLC, ≥99.9%), dichloromethane (puriss. ≥99%, GC), DPPH (2,2-diphenyl-1-picryl-hydrazyl-hydrate, for analysis), Trolox (6-hydroxy-2,5,7,8-tetramethylchroman-2-carboxylic acid, for analysis), acetic acid (glacial, ACS reagent, ≥99.7%), sodium acetate (ACS reagent, ≥99.0%), TPTZ (2,4,6-tris(2-yridyl)-s-triazine, ≥98.0%, HPLC), hydrochloric acid (ACS reagent, 37%), iron (III) chloride hexahydrate (ACS reagent, 97%), Folin–Ciocalteu reagent (for analysis), sodium carbonate (ACS reagent, anhydrous, ≥99.5%), gallic acid (ACS reagent, ≥98.0%), sodium nitrate (ACS reagent, ≥99.0%), sodium hydroxide (reagent grade, 97%), and quercetin dihydrate (grade ≥95%) were purchased from Millipore Sigma (Saint Louis, MO, USA).

### 3.2. Conventional Extraction

#### 3.2.1. Soxhlet Extraction

*D. salina* powder (10 g) was extracted with 300 mL of water, refluxed in Soxhlet apparatus for 18 h. The obtained extract solution was cooled to room temperature, centrifuged at 8000 rpm (15,740× *g*) for 20 min using a bench cooling centrifuge (Z 383 K, Hermle Labortechnik GmbH, Wehingen, Germany), and filtered through Whatman no.1 filter paper (Whatman™, Maidstone, UK) in order to remove the insoluble particles The water from the resulting solution was removed using a lyophilizer (VaCo 2-E, Zirbus technology GmbH, Harz, Germany). Dried extracts were stored under vacuum at −20 °C until further use.

#### 3.2.2. Maceration

Briefly, 5 g of *D. salina* powder was mixed with 150 mL of water and agitated at a moderate speed at room temperature for 24 h using a magnetic stirrer. Upon the completion of the extraction procedure, stirring was stopped; the extract was centrifuged at 8000 rpm (15,740× *g*) for 20 min using a bench cooling centrifuge (Z 383 K, Hermle Labortechnik GmbH, Wehingen, Germany) and filtered through Whatman no.1 filter paper (Whatman™, Maidstone, UK) in order to remove the insoluble particles. The water from the resulting solution was removed using a lyophilizer (VaCo 2-E, Zirbus technology GmbH, Harz, Germany) and the samples were stored at −20 °C for the subsequent analysis.

### 3.3. Ultrasound-Assisted Extraction

The extraction experiments were conducted using an Ultrasonic Processor UP200Ht (Hielscher Ultrasonics, Teltow, Germany), measuring 300 mm × 190 mm × 90 mm, operated at 26 kHz, with a rated power of 200 W and equipped with a sonotrode S26 d1 probe. For the operation parameters, the intensity was set at 100%, solvent composition (water at 100%), sample to solvent ratios *w*/*v* (1:10 and 1:5), extraction times (10, 20, and 30 min), and sonication mode (continuous, 0 s:0 s, or pulsed, 10 s:10 s) and tested. Water was chosen because it is a sustainable solvent, with no toxicity, and extraction occurred at room temperature (r.t., 24 °C). To avoid overheating and the consequent degradation of thermo-sensitive compounds, the experiments were performed in an ice bath. The extracts were centrifuged at 8000 rpm (15,740× *g*) for 20 min using a bench cooling centrifuge (Z 383 K, Hermle Labortechnik GmbH, Wehingen, Germany) and filtered through Whatman no.1 filter paper (Whatman™, Maidstone, UK) in order to remove the insoluble particles. All the extractions were performed twice and the yield of extraction (extractable components), expressed on a dry weight basis, was calculated from the following equation: Yield (g/100 g) = (w1 × 100)/w2, where w1 is the weight of the extract residue obtained after solvent removal and w2 the weight of the biomass before extraction.

Before each extraction, all samples were hand-homogenized, and then the tip probe was immersed in half of the total solvent height (4.5 cm). All extractions were performed at room temperature; however, samples were placed in ice to avoid overheating (and the consequent degradation of bioactives).

Extractions were performed in water and then microalgae suspensions were centrifuged (1118× *g* for 20 min) and extracts collected and stored under darkness at −4 °C for further analysis. The remaining pellet was dried at 60 °C in an oven until reaching a constant weight. A combination of different duty cycles (expressed as %) were applied, i.e., 100%, for a total of 10, 20, and 30 min as extraction times; the total cycle time comprised a pulse duration and a pulse interval. The amplitude (expressed as %) was also applied in 100%. The amplitude percentage refers to the percentage of maximum power used for the equipment.

### 3.4. Nutritional Composition

The general nutritional composition included the determination of moisture, ash, minerals, protein, total fat, and total carbohydrates. The determination of each parameter was performed in triplicate and the data were presented as mean ± SD. The moisture content of samples was measured gravimetrically through an automatic moisture analyzer PMB 202 (Adam Equipment, Oxford, NJ, USA) at 130 °C to a constant weight. The total ash content was determined by incineration at 500 °C in a muffle furnace [32]. Fat content was determined following the Portuguese standard method NP4168 [33]. Protein content (N × 6.25) was estimated by the combustion method DUMAS [34], using a Vario EL elemental analyzer (Elementar, Langenselbold, Germany). The carbohydrate content was calculated by the difference between the protein, fat, ash, and moisture contents.

### 3.5. Antioxidant Capacity Evaluation

#### 3.5.1. FRAP

The reducing power of the *D. salina* samples (raw water extracts and biomass fraction) was determined using the ferric ion-reducing antioxidant power (FRAP) assay [35]. The FRAP reagent was prepared by mixing 10 mmol/L 2,4,6-tripyridyl-s-triazine with 40 mmol/L HCl, 0.02 mol/L FeCl_3_ and acetate buffer, pH 3.6, in a ratio of 1:1:10. The *D. salina* samples (10 µL) were added to 290 μL of the FRAP reagent and the absorbance was measured at 593 nm after 6 min. Three replicates were performed for each sample, and the mean values of reducing power were reported as milligrams of Trolox equivalents per gram of dry weight (dw) and corresponded to the mean value of the triplicate tests.

#### 3.5.2. DPPH

The scavenging effect of raw *D. salina* and correspondent water extracts and biomass fractions was determined using the DPPH (2,2-diphenyl-1-picryl-hydrazyl-hydrate) methodology [36]. Aliquots of 10 μL of Trolox or *D. salina* samples were added to 100 μL (90 μmol/L) of the DPPH solution in methanol, and the mixture was diluted with 190 μL of methanol. In the control, the extract was substituted with the same volume of solvent, and in the blank probe, only methanol (290 μL) and the *D. salina* sample (10 μL) were mixed. After 30 min, the absorbance was measured at 515 nm. Three replicates were performed for each sample, and the mean values of the antioxidant capacity were reported as milligrams of Trolox equivalents per gram of dw and corresponded to the mean value of the triplicate tests.

### 3.6. Total Phenolic Content

The total phenolic contents (TPCs) of raw biomass and water extracts were evaluated using the method reported by [37]. Aliquots of raw *D. salina*, biomass, and extracts or gallic acid (30 μL) were added to 150 μL of 0.1 mol/L Folin–Ciocalteu reagent and mixed with 120 μL of sodium carbonate (7.5%) after 10 min. The mixtures were incubated in a dark at room temperature for 2 h, and then the absorbance was measured at 760 nm. The TPC was reported as milligrams of gallic acid equivalents per gram of dw and corresponded to the mean value of the triplicate tests.

### 3.7. Sensory Evaluation

Sensory analysis was conducted in a standardized sensory test room with booths, following the EN ISO 8589:2007 procedure. An untrained panel (*n* = 30; gender: females 17, males 13; age range: 22–47 years old) participated in the hedonic evaluation following the commonly used protocol by LEAF [38,39] in accordance with the ethical standards of the local committee responsible for human experiments and with the code of ethics of the World Medical Association [40]. Samples were randomly distributed, and the panelists were invited to sufficiently cleanse their palates with apples between trying the samples and pause at least 10 s between sniffs to partially restore the olfactory receptors. In addition to the control, 12 samples of extracts and 12 samples of biomass were offered to the panel in groups of three, and the evaluations occurred on different days. The panelists: (i) judged the level of odor intensity on a 6-point hedonic scale from no odor (0) to very strong odor (5), converted it into a percentage (%), and (ii) identified the odors as floral, citrus, sea/fish, earthy, and none of the abovementioned.

### 3.8. Statistical Analysis

The one-way analysis of variance (ANOVA), Tukey’s HSD test, Tukey’s multiple comparison test, Pearson’s correlation coefficients, agglomerative hierarchical clustering (AHC), and principal component analysis (PCA) were applied using Origin Statistical Software for Excel version 2021.4.1 (Addinsoft, New York, NY, USA) integrated with Microsoft Excel 2021 (Microsoft Corp., Redmond, WA, USA). A level of *p* ≤ 0.05 was considered as significant.

## 4. Conclusions

Ultrasound was applied as a highly effective, safe, and “green” cell disruption technology in microalgal biorefining. Effective cell disruption requires the careful selection of an appropriate ultrasonic frequency, intensity, and duration. The effects of the solid to solvent ratio, extraction time, and sonication mode (continuous or pulsed) were evaluated. Among the examined variables, the solvent concentration and time of extraction were found to be the most influential parameters, significantly affecting the extraction efficiency of protein and fat. The antioxidant capacity showed the same trend as the phenolic content, increasing with an increase in the solid to solvent ratio

Indeed, the measurement of target product release, coupled with other evaluation techniques, can offer a comprehensive and profound assessment of the extent of cell disruption. Together, these evaluation methods enable researchers to achieve a thorough understanding of the cell disruption’s impact on the target products, resulting in informed decisions concerning process optimization and better product yields.

UAE also had a positive impact on improving the sensory profile (odor) of the microalgal biomass. While the sea/fish odor became less intense, odors, such as earthy and citrus, became more intense, providing a better overall sensory profile score. UAE allowed us to improve the sensory profile of microalgae without losing their bioactive properties. To our knowledge, this work is the first to report the results of using UAE to improve the aroma profile of microalgae.

## Figures and Tables

**Figure 1 marinedrugs-21-00472-f001:**
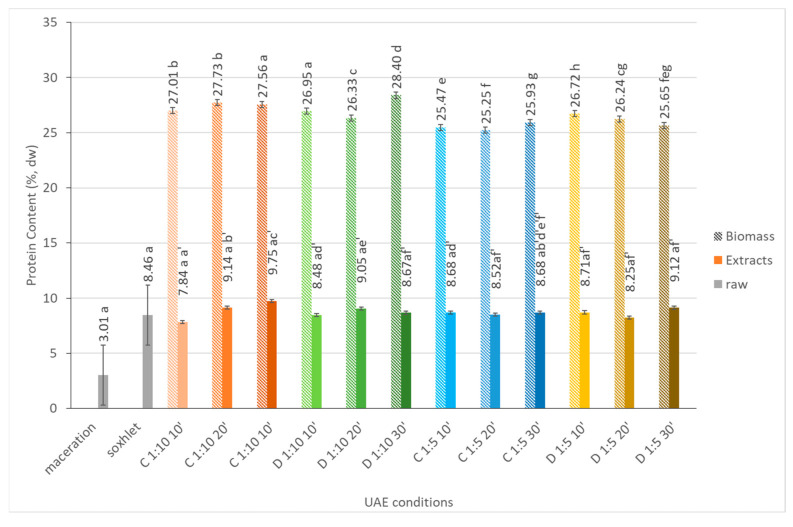
Protein content (%, dw) of *Dunaliella salina* raw, biomass, and extract fractions obtained under different ultrasound-assisted extraction conditions, using water as the medium. C: continuous sonication mode; D: pulsed sonication mode. Raw (gray solid fill), biomass (pattern fill), extracts (solid fill). The data shown are mean values (*n* = 3) followed by a letter. Different letters mean significantly different results (Tukey’s HSD; *p* ≤ 0.05).

**Figure 2 marinedrugs-21-00472-f002:**
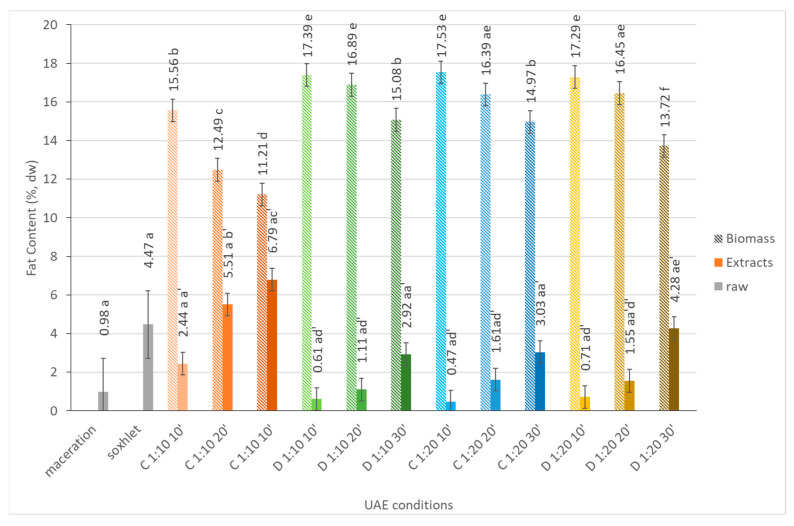
Fat content (%, dw) of *Dunaliella salina* biomass and extract fractions obtained under different ultrasound-assisted extraction conditions, using water as the medium. C: continuous sonication mode; D: pulsed sonication mode. Raw (gray solid fill), biomass (pattern fill), extracts (solid fill). The data shown are mean values (*n* = 3) followed by a letter. Different letters mean significantly different results (Tukey’s HSD; *p* ≤ 0.05).

**Figure 3 marinedrugs-21-00472-f003:**
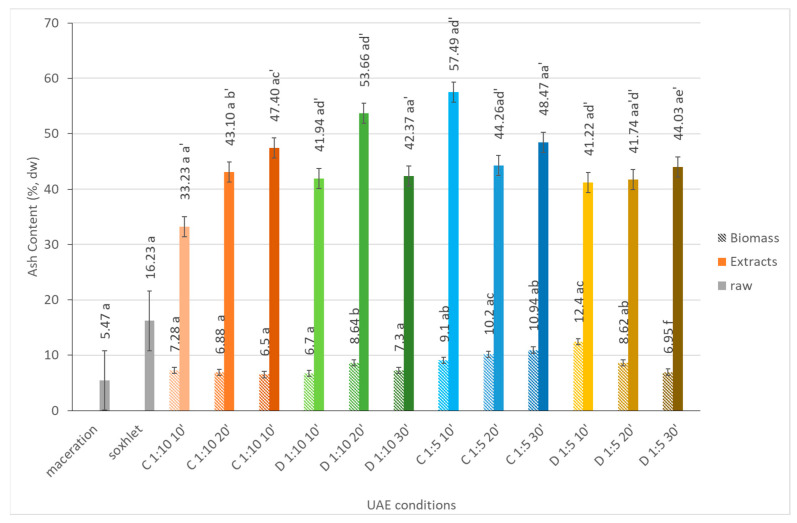
Ash content (%, dw) of *Dunaliella salina* biomass and extract fractions obtained under different ultrasound-assisted extraction conditions, using water as the medium. C: continuous sonication mode; D: pulsed sonication mode. Raw (gray solid fill), biomass (pattern fill), extracts (solid fill). The data shown are mean values (*n* = 3) followed by a letter. Different letters mean significantly different results (Tukey’s HSD; *p* ≤ 0.05).

**Figure 4 marinedrugs-21-00472-f004:**
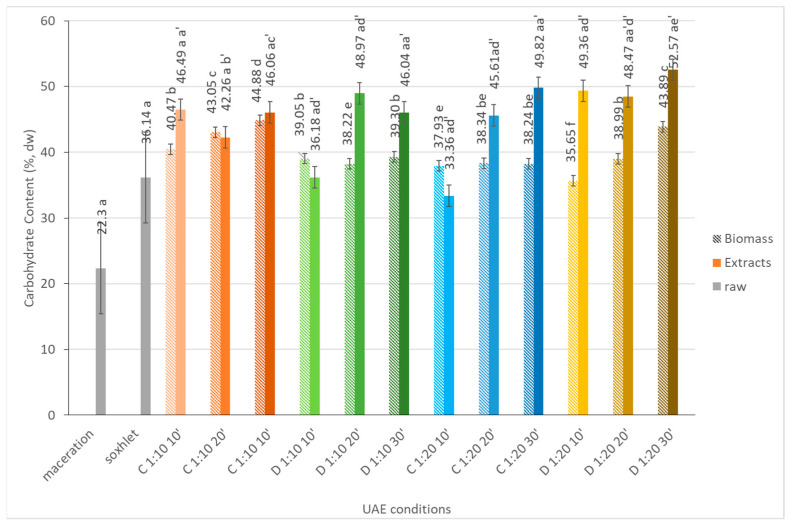
Carbohydrate content (%, dw) of *Dunaliella salina* biomass and extract fractions obtained under different ultrasound-assisted extraction conditions, using water as the medium. C: continuous sonication mode; D: pulsed sonication mode. Raw (gray solid fill), biomass (pattern fill), extracts (solid fill). The data shown are mean values (*n* = 3) followed by a letter. Different letters mean significantly different results (Tukey’s HSD; *p* ≤ 0.05).

**Figure 5 marinedrugs-21-00472-f005:**
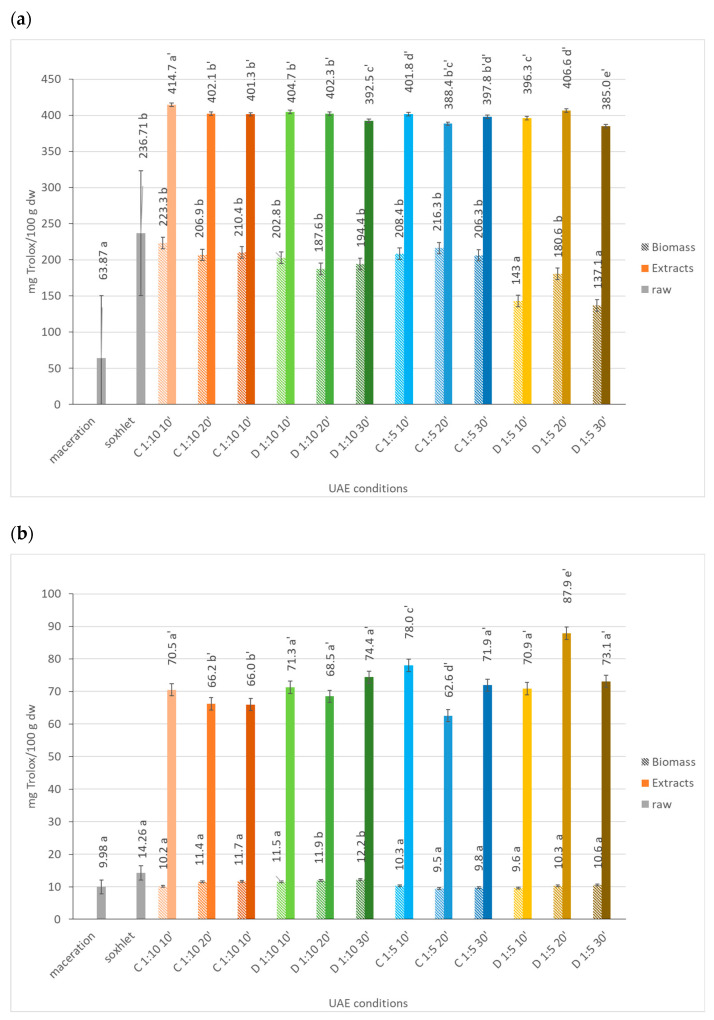
Antioxidant capacity of *Dunaliella salina* biomass and extract fractions obtained under different ultrasound-assisted extraction conditions, using water as the medium, measured by DPPH (**a**) and FRAP (**b**) assays. C: continuous sonication mode; D: pulsed sonication mode. Raw (gray solid fill), biomass (pattern fill), extracts (solid fill). The data shown are mean values (*n* = 3) followed by a letter. Different letters mean significantly different results (Tukey’s HSD; *p* ≤ 0.05).

**Figure 6 marinedrugs-21-00472-f006:**
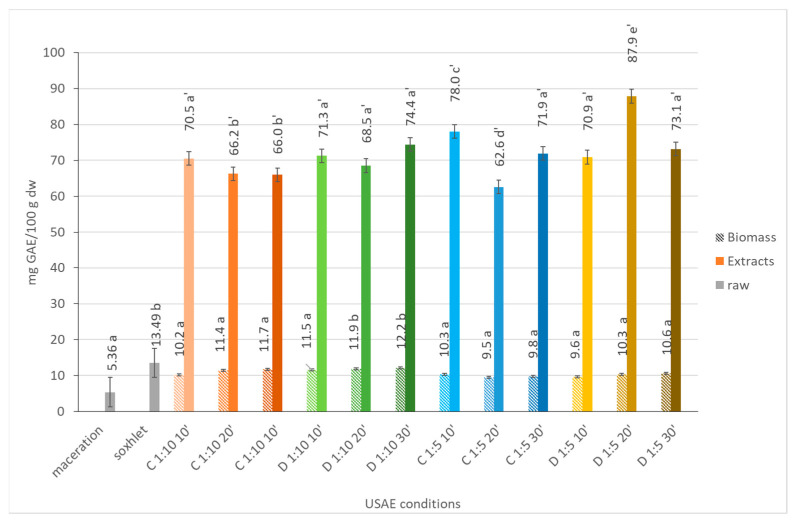
Total phenolic content (TPC), in mg GA/100 g dw, of *Dunaliella salina* biomass and extract fractions obtained under different ultrasound-assisted extraction conditions, using water as the medium. C: continuous sonication mode; D: pulsed sonication mode. Raw (gray solid fill), biomass (pattern fill), extracts (solid fill). The data shown are mean values (*n* = 3) followed by a letter. Different letters mean significantly different results (Tukey’s HSD; *p* ≤ 0.05).

**Figure 7 marinedrugs-21-00472-f007:**
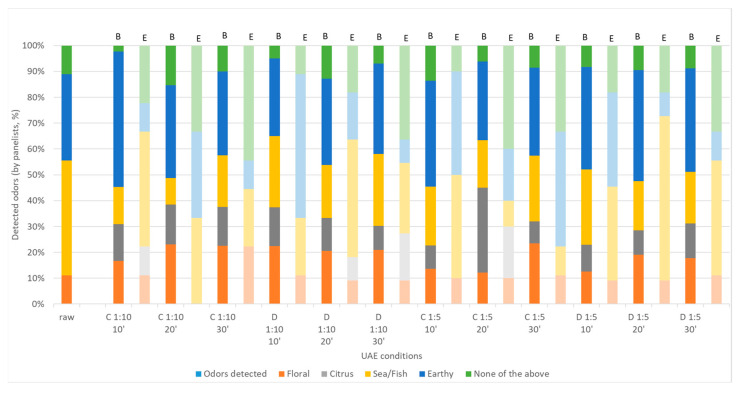
Detected odors by sensory evaluation (*n* = 30) in *Dunaliella salina* biomass (B: darker color) and extract (E: lighter color) fractions obtained under different UAE conditions, using water as the medium. C: continuous sonication mode; D: pulsed sonication mode.

**Figure 8 marinedrugs-21-00472-f008:**
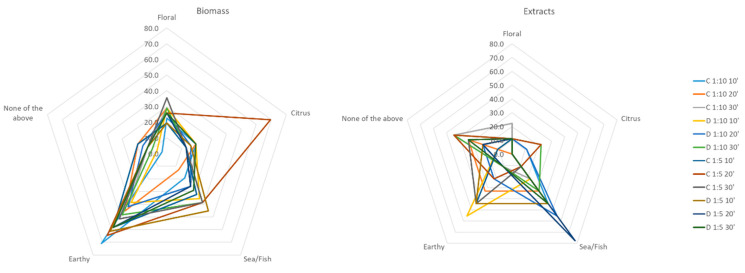
Sensory evaluation results (*n* = 30) of raw *Dunaliella salina* and correspondent biomass and extract fractions obtained after UAE using water as the medium.

**Table 1 marinedrugs-21-00472-t001:** Yields of recovery of the bioactive compounds from *Dunaliella salina* microalgal biomass using ultrasound-assisted extraction conditions, Soxhlet extraction, and maceration, using water as medium.

		Yield of Extraction (%)	
Solid to Solvent Ratio	Time (min)	Continuous Mode	Pulse Mode
1:10	10	35.78	22.82
	20	35.91	33.36
	30	41.38	35.30
1:5	10	23.52	14.85
	20	15.78	14.73
	30	17.47	14.67
**Conventional extraction**	**Time (h)**	**Yield of extraction (%)**	
Soxhlet	18	10.36	
Maceration	24	3.14	

## Data Availability

Data supporting the findings of this study are available upon request from the corresponding author.

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
