# Peer review of "Tuning the Bioactive Properties of Dunaliella salina Water Extracts by Ultrasound-Assisted Extraction"

_marinedrugs, 2023, doi:10.3390/md21090472_

Round 1

Reviewer 1 Report

In this work, the authors investigated how solid to solvent ratio, extraction time and sonication mode (continuous or pulsed) affect nutritional, sensory and antioxidant qualities of ultrasound assisted aqueous extracts of the microalgae Dunaliella salina.

Extraction of proteins and fats were most affected by solvent concentration and extraction time. Antioxidant capacity and phenolic content increased with an increase of solid to solvent ratio. Ultrasound extraction also improved the sensory profile, by diminishing fishy/sea-like odors in favor of citrus and earthy ones.

This is a listing of issues, some minor and some more serious, that need to be addressed before final publication:

Line 13: Instead of "Correspondence" it should read "Abstract"

Lines 14 and 38: "disruption" or "rupture"? 

Line 22: TO a small extent

Line 31: comma missing after "antioxidants"

Lines 49-50: on the one hand ... on the other hand

Line 59: nutritionally (adverb needed, not adjective)

Section 2.1 title reads "Samples and Reagents" 

However, I do not see any information on samples in this section, only about reagents. Please add the necessary details!

Also, for all reagents, please specify purity or at least reagent grade (HPLC-grade, spectroscopy grade, for analysis etc.)

Line 68: "quercetin dehydrate" - I do not understand. Is it "dehydrated quercetin" or "quercetin dihydrate"?

In the experimental, specify brand/version for all equipments: lyophilizer, centrifuge, freezer, furnace etc.

Specify vendor for all consumable (e.g. Whatman no. 1 filter paper)

Line 79: it is good practice to indicate also the g force of the centrifuge, as "8000 rpm" is relative to g force of the specific centrifuge employed

Line 88: because IT is

Lines 89-91: This part is confusing me. Were extractions performed at 24 degrees C or in an ice-bath (zero degrees)? Which one is true? Again confusing on lines 104-105. Figure 1 shows samples placed in ice bath during ultrasonic extraction, so how can that be room temperature extraction?

Lines 126 and 134-135: reference citations are not in MDPI style (use number-based referral)

For all chemical formulas, please make sure stoichiometry is indicated properly, i.e. using subscripts for atomic ratios e.g. FeCl3

Line 154: contents (plural)

Line 163: the word "standard" is repetitive in this phrase and anyways it is redundant with the "ISO" acronym. Cross out this word!

Line 169: Is cleansing the palate with "apples" standard protocol in such sensory analyses? Can you provide references to other sources in the literature that have used this protocol?

Revise the title of subsection 3.1.1 to resemble those of other subsections

Lines 185-186:  for (not "during") 18 hours

Line 192: allowed obtaining... (avoid using "us" and other first person constructs where possible; maximize impersonal writing style as much as possible)

Table 2 title must be revised: the table includes not only USAE conditions, but also yields obtained, so the title should reflect this

Line 372: "seem" (plural verb)

Lines 376-377: avoid using contracted forms (like "didn't") in scientific writing

Line 451: incredibly (adverb, rather than adjective, is needed here), although I would advise against using such superlatives when describing your data

There are several issues with Figure 8. First, it is not made explicit in the figure caption or the figure legend which bars (intensely colored or faded out) are for biomass and which are for extracts. Second, in the legend underneath the x-axis: "Odors detected" attributed to a pale blue color. What does that mean? (that some evaluators did not detect odor at all and there is only a limited percentage of participants that detected odors?) Where is this pale blue component in the chart?

The experimental part described some metal analyses using ICP-OES and cited a reference for the detailed protocol. Where are the results of these analyses? I could not find them in the current version of the manuscript.

The English is okay for the most part. Minor glitches here and there, which can be addressed by the authors themselves, without the need of professional help.

Reviewer 2 Report

The manuscript " Tuning the bioactive properties of Dunaliella salina water extracts by ultrasound-assisted extraction", reports relevant information on compounds polyphenols of Dunaliella salina with antioxidant properties. However, the authors should clarify the following points:

·        How they solved the filtration step required for ultrasound-assisted extraction

·        With this novel technique, it is possible to obtain compounds degraded by the high frequencies used. In this sense, could some of the reported compounds be degradation products?

·        The reported compounds were identified as glycosides or aglycones. Could the frequencies used have broken the O-sugar bond?

·        An HPLC or NMR experiment must be added to determine which compounds are present in the extracts. Likewise, through these spectra we will be able to determine the proportions, denaturation's, and types of polyphenolic compounds that exist after being subjected to different extraction methods.

·        The addition of chemical structures would help to understand the results.

The authors must correct the manuscript, since there are parts written in British English and parts written in American English.

Reviewer 3 Report

The comments are as follows:

1. More key-finding should be included in the Abstract.

2. I suggest using UAE as abbreviations for ultrasound-assisted extraction as the more commonly used.

3. Why did the authors choose to analyze microalgae Dunaliella salina? Why is its examination important? The explanation should be included in the Introduction section.

4. Are there any previous studies related to the characterization of this microalgae? Explain in the Intro. In that sense authors should emphasize the innovation of this study.

5. The authors should avoid using the term "optimisation" unless they have done RSM model or statistical optimisation model.

6. Figure 1 should be removed since the setup is well-known.

7. The text parts from line 99-101, 104-106 and 112-113 should be removed since that is a repetiton.

8. Table 2 should be removed since it is a repetiton. All extraction conditions are mentioned in the text.

9. SD values are missing in Table 2.

10. The authors should revise the text for typos.

11. The authors should avoid using the words like we, our etc.

12. Lines 216-219. How the authors explain the obtained results and the decrease in yield with increasing extraction time?

13. Lines 229-232. The finding from these previous studies are not comparable to this study because temperature was not held constant as is the case in this study.

14. Figures 2-7. There is no clear difference in labeling between raw and biomass. It should be improved.

15. The abbreviations should be defined just the first time they are mentioned in the text and used as such throughout the text.

16. A lot of repetitions in the discussion of the results.

17. Lines 372-374 and 379-380. The temperature is kept constant (24oC) in the UAE, isn't it?

18. What is the purpose and practical application of the finding related to the sensory analysis of UAE extracts?

19. The conclusion should be supported by more specific results. Overall, what are the optimal conditions of the UAE?

20. The authors should cite recent studies.

Round 2

Reviewer 3 Report

The authors should approach the improvement of the quality of the work in much more detail and in accordance with the previous comments. Overall, the manuscript needs a much more thorough analysis and better interpretation and explanation of the results supported by facts.
